# When Vision Fails:
# Text Attacks Against ViT and OCR

## Abstract

Text-based machine learning models are vulnerable to an emerging class of Unicode-based adversarial examples capable of tricking a model into misreading text with potentially disastrous effects. The primary existing defense against these attacks is to preprocess potentially malicious text inputs using optical character recognition (OCR). In theory, OCR models will ignore any malicious Unicode characters and will extract the visually correct input to be fed to the model. In this work, we show that these visual defenses fail to prevent this type of attack. We use a genetic algorithm to generate visual adversarial examples (i.e., OCR outputs) in a black-box setting, demonstrating a highly effective novel attack that substantially reduces the accuracy of OCR and other visual models. Specifically, we use the Unicode functionality of combining characters (e.g., $\tilde{n}$ which combines the characters $n$ and $\sim$) to manipulate text inputs so that small visual perturbations appear when the text is displayed. We demonstrate the effectiveness of these attacks in the real world by creating adversarial examples against production models published by Meta, Microsoft, IBM, and Google. We additionally conduct a user study to establish that the model-fooling adversarial examples do not affect human comprehension of the text, showing that language models are uniquely vulnerable to this type of text attack.

## 1 Introduction

Adversarial examples are inputs to models that contain subtle perturbations designed to cause models to output undesirable results. They were first identified in the image domain, where small changes in pixel values could cause classifiers to fail (Szegedy et al., 2014; Biggio et al., 2013; Goodfellow et al., 2014). Later advances brought adversarial examples to natural language, where text encoded as Unicode (The Unicode Consortium, 2021a) could have characters inserted or replaced to create adversarial examples in the text domain (Boucher et al., 2022; Pajola & Conti, 2021).

Until now, it has been thought that vision model designed to process text were robust against Unicode adversarial examples. Model architectures such as Vision Transformers (ViTs) (Salesky et al., 2021) and Optical Character Recognition (OCR) pre-processing (Boucher et al., 2022) leverage rendered text as input, suggesting that adversarial perturbations at the text encoding level would not affect these models. We show that this is not the case.

In this work, we show that these visual models are themselves vulnerable to Unicode-based attacks. We propose a technique to encode Unicode perturbations in a way that, once rendered, the image of the rendered text will contain adversarial perturbations that defeat visual defenses. By leveraging combining marks from the Unicode specification, we can craft small, targeted visual perturbations that appear on the rendered image of text. While these marks are not visually significant enough to impact a human reader's understanding of text (as we show in a user study), the manipulated pixels in the image domain of rendered text enable targeted attacks on model outputs. This class of adversarial examples is distinct from general image adversarial examples in that pixel values are not perturbed directly, but rather are modified by perturbing pre-rendered Unicode.

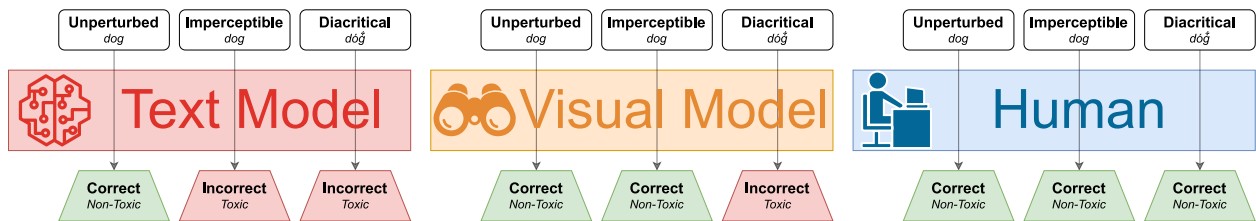

Figure 1: The visualization gap, against which visual models claim to be robust, continues to be a source of adversarial examples via diacritics as in this toxic content classifier example.

In particular, we build on existing adversarial text perturbation techniques to introduce a new class of Unicode-based adversarial examples that leverages Unicode's combining diacritical mark functionality. Prior work claiming ViT and OCR are robust against text-based perturbations (Boucher et al., 2022; Salesky et al., 2021) did not consider text containing combining diacritics prior to visual rendering, which we leverage in this work to demonstrate OCR's vulnerability to Unicode-based attacks. We demonstrate that these barely perceptible marks have a substantial impact on ViT and OCR model performance, causing performance drops of around 60% to as high as 92.6%.

Text-based attacks on ViT and OCR have broad implications, particularly in cases involving large-scale, automated text input with minimal human review. For example, we demonstrate extensions to attacks on toxic content detection (Hosseini et al., 2017) and machine translation (Belinkov & Bisk, 2018) that succeed despite ViT and OCR defenses against encoding attacks. The techniques we propose can even be used to affect printed text, such as addresses on envelopes sent through the U.S. Postal Service (which uses OCR to speed up processing (United States Postal Service, 2023), or disclosures submitted as part of a legal case. In the latter example, if the disclosing legal team adds targeted text perturbations to printed disclosure documents, the text can form visual adversarial examples that are incorrectly processed by OCR systems despite being readable by humans. Large-scale document scanning in particular is often automated and performed with little to no human review of the original paper documents—and so it is entirely plausible that a system may scan encoded text (and therefore produce incorrect output) without detection. This could, for instance, prevent searchability, thus impeding the opposing legal team in the discovery process.

In summary, we make the following contributions in this paper:

- We introduce a novel form of adversarial example against machine-learning models that process rendered text, including Vision Transformers and Optical Character Recognition models, which are encoded in the textual domain but operate in the visual domain.

- We demonstrate that existing defenses (Boucher et al., 2022; Salesky et al., 2021; Clark et al., 2022) for encoding attacks against text models including ViT, OCR, and neural encoders are insufficient.

- We conduct a user study to validate that our adversarial examples do not impact human readability and comprehension.

- We propose definitive defenses for diacritical attacks against Unicode-based visual models.

## 2 Background

### 2.1 The Visualization Gap

Traditionally, machine learning models processing text such as natural language operate directly upon some encoding of the input text. This may take the form of input embeddings as vectors representing words, characters, or learned subword components created by parsing Unicode inputs (The Unicode Consortium, 2021a). However, unlike models, humans do not directly consume encoded text. Rather, text is rendered and then visually presented to human users.

It is here that a security design flaw arises: the relationship between encoded text and rendered text is not bijective. That is, a visual rendering could be represented by many unique text encodings. Formally,

$$\forall t \in T, \quad U(t) \nLeftrightarrow \{v(t)\} \tag{1}$$

where $T$ is the set of all possible text sequences, $U$ is the function generating the set of all possible Unicode representations of a text, and $v$ is the visual rendering of a text.

Consider, for example, the presence of invisible characters such as Unicode's Zero-Width Space (ZWSP); these characters have no effect on the rendering of most text and yet change the encoded representation. Visually-identical characters, known as homoglyphs, can also be used interchangeably, and control characters can be used to delete and reorder characters.

The difference between the encoding and visualization of text can be used to create adversarial examples against models that operate directly upon some form of textual input (Boucher et al., 2022; Pajola & Conti, 2021), improving the stealth of earlier techniques leveraging misspelling or paraphrasing (Gao et al., 2018; Li et al., 2018; Belinkov & Bisk, 2017; Khayrallah & Koehn, 2018). The visualization gap is depicted in Figure 1.

## 2.2  Defense Through Vision

To defend against adversarial examples that exploit the visualization gap, model designers must seek to unify the text encoding and visualization pipelines. That is, designers must seek to build or augment models such that:

$$\forall t \in T, \quad E(U(t)) = \{t'\} \Leftrightarrow \{v(t)\} \tag{2}$$

where $E$ generates the set of embeddings for the encoded values taken as input.

One simple but effective way to accomplish this on existing models is to render text inputs and process the resulting images through OCR as a pre-processing step prior to model inference (Boucher et al., 2022). In effect, this provides an automated system that maps fixed visual renderings to a common encoded input. The inference pipeline in this setting is: *encoded input → rendered image → text → model*.

For greenfield models, Vision Transformers may be the preferred defense approach as no compute-intensive pre-processing model is required. ViTs operate upon images as input, and operating directly upon rendered images as embeddings yields both good performance and defense-by-design against attacks exploiting the visualization gap (Salesky et al., 2021). The inference pipeline in this setting is: *encoded input → rendered image → model*.

Finally, neural encoders offer new NLP models some robustness against Unicode perturbations (Clark et al., 2022). Although they do not operate in the visual domain, neural encoders are a form of learned embedding that map relationships between Unicode characters such that encoded values that would look similar if rendered should result in similar embeddings. The inference pipeline in this setting is: *encoded input → neural embedding → model*.

## 2.3  Diacritical Marks

Diacritics – also known as diacritical marks or accents – are small marks that can be placed on top of other written letters. These marks serve different purposes across different linguistic families, but often serve to modify the pronunciation or meaning of words. Examples of diacritics include the Spanish ñ, the German ä, and the French è. Diacritics are most common in European languages, but are also used to aid in the romanization of non-Latin scripts such as the case of pinyin for Mandarin.

In Unicode, characters commonly used with diacritical marks typically have a dedicated character – or code point – in the Unicode specification. However, the concept of *combining diacritical marks* also exists. These characters modify the character immediately preceding them to add the specified diacritical mark. The Unicode specification defines 256 such combining characters (The Unicode Consortium, 2021b;d;f;e;c).

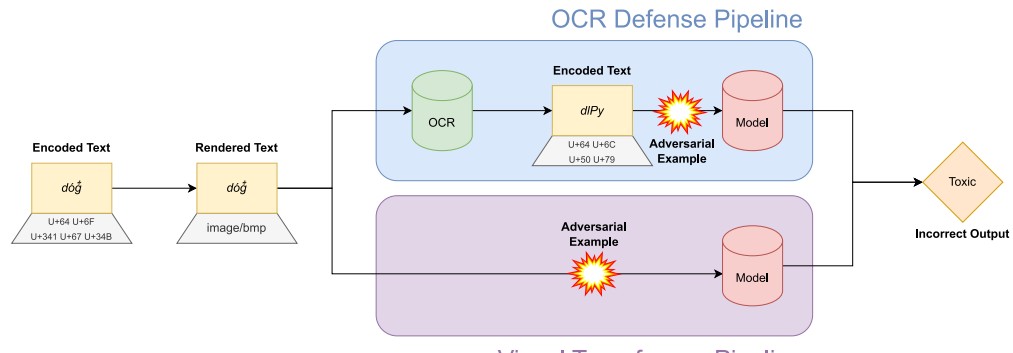

Figure 2: Our attack: adversarial examples in the visual domain encoded in the textual domain.

## 3 Related Work

### 3.1 Text-Based Adversarial Examples

While most adversarial examples proposed in prior work have been image classification attacks (Szegedy et al., 2014; Goodfellow et al., 2014; Athalye et al., 2018), recent work has demonstrated the existence of adversarial examples against text-based ML models. In particular, the accuracy of text classification models decreases on noisy inputs and it is possible to deliberately craft adversarial texts targeting systems reliant on text understanding (Gao et al., 2018; Li et al., 2018; Belinkov & Bisk, 2017; Khayrallah & Koehn, 2018; Alzantot et al., 2018; Zou et al., 2019).

In contrast to prior work presenting visibly perceptible perturbations, Boucher et al. (2022) previously showed that the discrepancy between text encoding and text visualization can be exploited as an attack vector against text-based machine learning models. They proposed optical character recognition (OCR) models as a "catch-all defense" against such attacks, arguing that the visual nature of the input would mitigate differences in visual and text encoding representation that left text-based processing systems vulnerable.

While Boucher et al. (2022) evaluated OCR as a defense against imperceptible perturbations, diacritics present a new class of attacks that are often scarcely more perceptible than invisible perturbations but have a substantial impact on OCR accuracy. Salesky et al. (2021) showed that Arabic diacritics and other minor text modifications substantially degrade the performance of text-based models, but concluded that ViTs achieve "relatively robust" performance against diacritized text. In this paper we demonstrate that both OCR systems and ViTs are indeed vulnerable to the Unicode-based adversarial examples introduced by Boucher et al. (2022), and that such examples significantly impact the performance of these models.

### 3.2 Attacks on OCR

Most prior work in breaking OCR systems has focused on visual CAPTCHAs (Yan & El Ahmad, 2007; Azad & Jain, 2013), but OCR is commonly used in a variety of preprocessing tasks. Recently, Chen et al. (2020) demonstrated a class of attacks on OCR using adversarial watermarks in a twist on classic adversarial image examples. The commonality of watermarks on inputs to OCR systems makes this attack effectively imperceptible to humans, though this attack relies on non-text image data in inputs. In contrast, in this work we present a class of attacks in which the adversarial examples are encoded directly into text rather than perturbing an image's background. To the best of our knowledge, ours is the first such attack that manipulates pre-rendered text directly, using the resulting image as an adversarial example.

### 3.3 Toxic Content Detection

Identifying toxic content online has proven to be one of the most pervasive yet elusive research questions in natural language processing. A rapidly growing area of research, toxic content detection is challenging

even when classifying standard Unicode characters in the English language, and toxic content classifiers are generally not robust against adversarial examples (Kurita et al., 2019; Risch & Krestel, 2020). Online users commonly deploy a variety of simple text modifications to bypass platform filters: Kurita et al. (2019) gives the adversarial example of writing "s*ut up" in the place of 'shut up'. While most adversarial examples against toxic content detection are visually perceptible, Boucher et al. (2022) showed it is also possible to modify the Unicode text encoding of toxic content in a way that is visually imperceptible but fools content detection models.

## 4 Methods

### 4.1 Exploiting Diacritics

In the image domain, adversarial examples are typically crafted by slightly perturbing the values of key pixels often identified through a gradient-based approach (Goodfellow et al., 2014). While such an approach would in theory work against ViTs and OCR models, the visual text domain has the added constraint that the input images are generated through rendering text. Therefore, it is not possible to arbitrarily perturb pixel values, as pixel values are not directly encoded; rather, the pixels are generated from encoded text.

Combining diacritical marks provide a method by which arbitrary characters can be subtly perturbed in the image domain. These marks will produce a noticeable visual effect, but as we will later demonstrate through a user study these marks do not affect a human's ability to read the text. However, just as changing particular pixel values can cause general image classifiers to fail, so too can the perturbed pixel values created by diacritical marks affect ViTs and OCR models. Although the size of the perturbation space is smaller than the more continuous space provided by color pixels in unconstrained images, we will later demonstrate through a series of experiments that this space is sufficiently large to generate adversarial examples for text. This attack is visualized for ViT and OCR pipelines in Figure 2.

### 4.2 Attack Technique

Visual adversarial examples encoded as text can be generated in a black-box model using a gradient-free optimization technique. We built our technique using differential evolution, a genetic optimization algorithm (Storn & Price, 1997). Adopting a similar approach previously proposed for imperceptible perturbations (Boucher et al., 2022; Shumailov et al., 2021), we use differential evolution to minimize similarity with the reference output for sequence-to-sequence tasks, and to minimize the output probability of the target class for classification tasks.

The algorithm to generate visual adversarial examples encoded as text is provided as Algorithm 1. In summary, this algorithm takes as input a string to perturb, the target output class or reference output sequence, a visual text model, a perturbation budget representing the maximum number of injected diacritics, and a set of diacritics from which to select. A set of optimization parameters specific to differential evolution are also required by the algorithm (Storn & Price, 1997), although reasonable defaults may be used across repeated invocations. We provide the parameters chosen for our implementation in the following section. In the case of classification models, this algorithm will return a version of the input string perturbed with diacritics up to the allowed budget that minimizes the output probability of the supplied target class. In the case of sequence-to-sequence models – such as the machine translation task – the algorithm minimizes the supplied similarity metric with the supplied reference output.

The threat model for this attack contains an adversary who is able to submit inputs to a model, either via an API or locally. The adversary does not have access to the model's weights (i.e. a black-box setting) and the target model leverages *vision-driven defenses* for encoding attacks such as OCR pre-processing or a ViT architecture. The adversary's goal is to craft an input to the model that will cause an incorrect output.

---

**Algorithm 1:** Black-box generation of visual text adversarial example

---

**Input:** text $\mathbf{x}$, reference output $\mathbf{y}$, model $\mathcal{M}$, perturbation budget $\beta$, diacritics list $\mathbf{D}$
**Optimization Params:** similarity metric $\mathcal{S}$, population size $s$, evolution iterations $m$, differential weight $F \in [0, 2]$, crossover probability $CR \in [0, 1]$
**Result:** Encoded text visually similar to $\mathbf{x}$ which is an adversarial example against $\mathcal{M}$ when rendered

**procedure** PERTURB $(p)$
  $\bar{x} := \mathbf{x}$
  **for** $n := 0$ **to** $\beta$ **do**
    $d, i := p_n$
    **if** round$(i) \geq 0$ **then**
      $\bar{x} = \text{insert}(\bar{x}, \mathbf{D}_d, \text{round}(i))$
    **end if**
  **end for**
  **return** $\bar{x}$
**end procedure**

Randomly initialize   $\mathbf{P} := \{\mathbf{p_0}, \dots, \mathbf{p_s}\}$,
       where $\mathbf{p_n} := [(d_0, i_0), \dots, (d_\beta, i_\beta)]$,
   where $d_n \sim \mathcal{U}(0, |\mathbf{D}|)$, $i_n \sim \mathcal{U}(-1, |\mathbf{x}|)$
            $\triangleright \mathcal{U}$ is uniform dist.

**if** $\mathcal{M}$ is classifier **then**
  $\mathcal{F}(\hat{x}) = \mathcal{M}(\hat{\mathbf{x}})$     $\triangleright$ logit of target class
**else**
  $\mathcal{F}(\hat{x}) = \mathcal{S}(\mathcal{M}(\hat{\mathbf{x}}), \mathbf{y})$
**end if**

**for** $i := 0$ **to** $m$ **do**
  **for** $j := 0$ **to** $s$ **do**
    $\mathbf{p_a}, \mathbf{p_b}, \mathbf{p_c} \xleftarrow{\text{rand}} \mathbf{P}$ s.t. $j \neq a \neq b \neq c$
    $R \sim \mathcal{U}(0, |\mathbf{p_j}|)$
    $\hat{\mathbf{p}}_\mathbf{j} := \mathbf{p_j}$
    **for** $k := 0$ **to** $|\mathbf{p_j}|$ **do**
      $r_j \sim \mathcal{U}(0, 1)$
      **if** $r_j < CR$ **or** $R = k$ **then**
        $\hat{\mathbf{p}}_{\mathbf{j_k}} = \mathbf{p_{a_k}} + F \times (\mathbf{p_{b_k}} - \mathbf{p_{b_k}})$
      **end if**
    **end for**
    **if** $\mathcal{F}(\text{PERTURB}(\hat{\mathbf{p}}_\mathbf{j})) < \mathcal{F}(\text{PERTURB}(\mathbf{p_j}))$ **then**
      $\mathbf{p_j} = \hat{\mathbf{p}}_\mathbf{j}$
    **end if**
  **end for**
**end for**
$\bar{\mathbf{f}} := \{\mathcal{F}(\text{PERTURB}(\mathbf{p_0})),$
      $\dots, \mathcal{F}(\text{PERTURB}(\mathbf{p_s}))\}$
**return** PERTURB$(\mathbf{P}_{\text{argmax}(\bar{\mathbf{f}})})$

---

## 5 Evaluation

The experimental evaluation of visual text adversarial examples requires examining two claims: first, that the adversarial examples effectively degrade the performance of a broad set of models, and second, that the adversarial examples do not affect human comprehension.

To examine the first claim, we will generate adversarial examples for an OCR model and measure performance degradation. We will then place this model into a pipeline that renders text input, performs OCR, and then calls the downstream model for both machine translation and toxic content detection and evaluate

Table 1: Model performance against adversarial examples of diacritical mark budget 5.

| Model | Baseline Perf | Adv. Ex. Perf $\beta = 5$ | Attack Perf Drop |
|---|---|---|---|
| TrOCR (Li et al., 2021) | 0 | 12 | n/a |
| TrOCR-FairSeq (Ott et al., 2019) | 60.9 | 24.2 | 60.3% |
| TrOCR-IBM Toxic (IBM, 2020) | 87.6 | 31.8 | 63.7% |
| ViT FairSeq (Salesky et al., 2021) | 57.9 | 24.5 | 57.7% |
| CANINE SQuAD (Clark et al., 2022) | 67.6 | 5.0 | 92.6% |

the performance of adversarial examples. Next, we will craft and measure adversarial examples for a ViT performing machine translation directly on rendered text. Finally, we will demonstrate an extension of these attacks to a neural encoder model performing question answering. A summary of the experimental results is given in Table 1. These results are presented over varying budgets in Figure 3, with additional metric-specific visualizations given in Figures 5 to 9 in the Appendix. We note that since the concept of visual text adversarial examples is new, there are no direct baseline metrics in the literature against which to compare these results.

All adversarial examples were generated on a cluster of machines each equipped with a Tesla P100 GPU and Intel Xeon Silver 4110 CPU running Ubuntu. We followed Algorithm 1 for example generation, selecting a a population size of 32, 10 iterations, a crossover probability of 0.7, a differential weight dithering from 0.5 to 1, and a varying budget ranging from 0 to 5. Where applicable, we select chrF (Popović, 2015) as the similarity metric $\mathcal{S}$. For each experiment, we generate 500 adversarial examples for each budget value. Perturbations are generated from the subset of diacritical marks in Unicode supported by the Microsoft Arial Unicode font (Microsoft, 2021), which is `U+300-U+346` and `U+360-U+361`. All datasets and models used are publicly available for research purposes. All experimental code and results are made available for future research at anonymous.4open.science/r/diacritics.

### 5.1 TrOCR

TrOCR is a transformer model (Vaswani et al., 2017) published by Microsoft implementing OCR (Li et al., 2021). TrOCR achieves state-of-the art performance on text recognition and leverages the modern transformer architecture.

Our first experiment evaluates the performance of TrOCR against diacritics injected via Algorithm 1. For each example, we render the input using the Microsoft Arial Unicode font (Microsoft, 2021) and pass the rendered image to the TrOCR model for inference. In this experiment, we use negative Levenshtein distance of the model output with the adversarial input as the similarity metric. If the TrOCR model is highly performant, we would expect - and indeed do see - a small average distance between the input and output when the attack budget is zero (representing no diacritic injections). The experiments seek to generate adversarial examples using diacritics injected into inputs sampled from the grammatically-correct validation split of the CoLA dataset (Warstadt et al., 2018), selected as a sample of short English-language inputs.

In Figure 5 we plot the distance between the output and both the perturbed and unperturbed input. Both measures grow with the budget, indicating that the model neither removes diacritical marks nor recognizes them as diacritics.

### 5.2 Defended FairSeq

In this set of experiments, we evaluated an English to French machine translation model published with Facebook's FairSeq toolkit (Ott et al., 2019). The model is a transformer architecture (Vaswani et al., 2017) trained on the WMT14 EN→FR corpus (Ott et al., 2018).

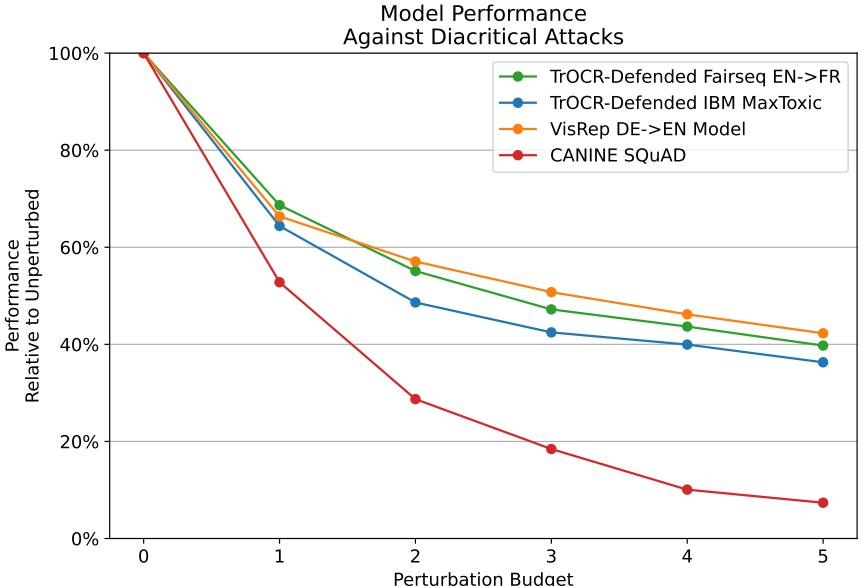

Figure 3: Performance of evaluated models across perturbation budgets relative to no perturbations.

Following the retrofitted defense tactic previously proposed for Unicode attacks (Boucher et al., 2022), we placed the FairSeq EN→FR model in a pipeline that first renders text input and performs OCR via TrOCR. The output of TrOCR is then passed as input to the FairSeq model. The experiments generate adversarial examples using diacritics for inputs drawn from the test split of the WMT14 EN→FR dataset.

In Figure 6 we plot the average chrF score (Popović, 2015) – a measure of translation quality selected as the similarity metric for this experiment – relative to the reference translation as computed by sacreBLEU (Post, 2018). The results indicate a decreasing translation quality with increasing diacritics budget.

### 5.3 Defended Toxic Classifier

Similarly, we evaluated another model defended from Unicode attacks via TrOCR. In this experiment, the downstream model is IBM's toxic content classifier (IBM, 2020). We use the Wikipedia Detox Dataset (Thain et al., 2017) as the inputs from which we craft adversarial examples.

The classification performance of the defended model over varying attack budgets is shown in Figure 7. There is, again, a negative relationship between the performance of the model and the attack budget. In fact, with a budget of just two diacritical marks, the performance of the model falls below random (i.e. 50% correct classification).

### 5.4 Visual FairSeq

In this experiment, we evaluate the performance of ViTs against diacritic attacks. Specifically, we target the fork of FairSeq (Ott et al., 2019) implementing German to English machine translation operating directly on visual inputs. This is the model used by Salesky et al. (2021) in their Visual Transformer proposal. Since this model operated upon inputs encoded as images, no OCR defense is necessary. We used the test split of the WMT20 DE→EN news dataset as to generate adversarial examples (Mathur et al., 2020).

In Figure 8 we plot the average chrF score over the perturbation budget. Similar to the TrOCR-defended machine translation experiment, we see a significant negative correlation between translation quality and diacritics perturbation budget.

### 5.5 CANINE Question Answering

Neural encoders are a recent NLP tool to replace dictionaries with embeddings learned directly from Unicode code points. Although these models do not unify the visual and encoded pipelines for text input, the learned embeddings claim to offer better flexibility for unknown characters. In this experiment, we evaluate whether diacritic attacks can subvert models leveraging neural encoders.

We evaluated a question answering model leveraging the CANINE neural encoder proposed by Google (Clark et al., 2022). Specifically, we evaluated a transformer model (Hsu, 2022) trained on the SQuAD dataset (Rajpurkar et al., 2016). Consistent with benchmarks against this dataset, we select F1 score as the similarity metric. The results, shown in Figure 9, show a sharp decline in F1 score with increasing diacritical perturbations.

## 6 User Study

Having shown that OCR and ViT models perform poorly when interpreting text containing visual perturbations and that these failures can be used to craft adversarial examples for downstream tasks, we conduct a user survey to understand how diacritical perturbations affect human comprehension.

### 6.1 User Study Methodology

The survey contained two high-level sections which are each shown in Appendix A.3.

First, to measure human ability to read text containing diacritics, we ask respondents to retype sentences containing diacritics, specifying that they should omit diacritics or other marks. We selected 5 short examples to measure comprehension.

Second, to measure human comprehension of text containing diacritics, we asked respondents to identify whether a short sentence is toxic using the toxicity definition of Google's Perspective API (Google, 2021). For the sentences with a ground-truth toxic label, we filtered to select milder language. We used 12 sentences in total, evenly split between six toxic and six non-toxic examples. The sentences were randomly ordered within the survey question to avoid biasing respondents.

We surveyed 200 people using Amazon Mechanical Turk, limiting the survey to include only workers with at least a U.S. high school education since our questions concerned critical interpretation of text. The survey was advertised as "Answer a short (estimated 3-4 min) survey about reading and interpreting short sentences", intentionally avoiding any mention of diacritics, accents, or unusual marks. We also warn potential respondents that the survey "contains language that may be considered rude or disrespectful" prior to accepting the task to ensure that they are appropriately informed about the survey contents.

In analyzing the results, we identify six identical low-quality responses that failed quality-control screening, leaving a total of $n = 194$ responses. We paid the remaining respondents \$1.50 USD for completing the survey, an amount chosen to reflect a fair hourly wage. The survey had an average completion time of just under three minutes and maximum completion time of up to ten minutes.

### 6.2 User Study Results

Overall, the results demonstrate that diacritical marks do not significantly affect a human user's ability to comprehend the adversarial examples generated in the prior experiments. User responses were highly accurate in both reading and interpreting text containing diacritical marks, indicating that OCR and ViT models struggle with a text interpretation task humans find simple.

#### 6.2.1 Text Legibility

We find that the overwhelming majority of human readers are able to correctly read and repeat the example sentences, with respondent accuracy rates ranging from 93.3% to 97.9%. In evaluating what constitutes an 'accurate' response, we ignore minor punctuation differences that seem unlikely to have been caused by

| Example | Ground Truth Toxicity Label | Model Label | Human Reader Label | Human Reader Performance | Model Correct | Humans Correct |
|---|---|---|---|---|---|---|
| Y̧ịur house is ūg̃ly, too. | True | False | True | 93.8% | ✗ | ✓ |
| I didn't ẙandalize your user page, ī̃diot. | True | False | True | 94.8% | ✗ | ✓ |
| ::Añ̄,yes, aẙ̈ ảsshole | True | False | True | 96.4% | ✗ | ✓ |
| Ahå ̄ f̥oung̊ it . − | False | True | False | 99.0% | ✗ | ✓ |
| :I'Îl̂ˆrep̊ly on youř talk page. | False | True | False | 99.0% | ✗ | ✓ |
| Pìç of t̲he day ̄Wednesday | False | True | False | 99.0% | ✗ | ✓ |
| thiś band sucks anyways | True | True | True | 80.4% | ✓ | ✓ |
| You are a˝ċoward a̲ňd a fool. | True | True | True | 97.9% | ✓ | ✓ |
| Your motheṙ is illiterate! | True | True | True | 93.3% | ✓ | ✓ |
| It¸is¸a golden hair. | False | False | False | 98.5% | ✓ | ✓ |
| It d̶id not rain. | False | False | False | 99.5% | ✓ | ✓ |
| John oft̲e̦n¨mee̦ts Mary. | False | False | False | 99.0% | ✓ | ✓ |

Table 2: Comparison of OCR and ViT model and human performance detecting toxicity in adversarial examples. 'True' means the adversarial example was considered toxic, while 'False' indicates the example was non-toxic.

diacritical marks (e.g., lack of capitalization, quotation marks around the sentence, a lack of a period at the end of the sentence, etc.).

Inaccurate responses mostly substituted a word that contained a diacritical mark for a similar but incorrect word (e.g., "tube" instead of "tub", or "John laid" instead of "John lay"). In such cases, we take a conservative approach and err on the side of marking it as incorrect since we cannot determine whether the substitution was due to the presence of diacritical marks or to ordinary typographical human errors. We conclude that while a small minority of respondents struggled to identify certain words containing diacritics, the vast majority of human readers are readily capable of comprehending text regardless of diacritical marks.

### 6.2.2 Text Comprehension

In the second question, we test individuals' ability to identify toxic text as a proxy for human comprehension of the text. Table 2 contains the 12 examples used in the survey, evenly divided between six examples the TrOCR toxic content classifier labeled correctly (based on the ground-truth toxicity label from the Wikipedia Detox Dataset) and six it did not. Human readers were able to correctly classify all 12 sentences, including the six examples that the TrOCR model classified incorrectly, with respondent accuracy ranging from 80.4% to 99.0% for a given sentence.

In some cases, the percentage of human readers identifying a sentence as toxic was seemingly affected more by individual perceptions of toxicity than by the diacritical marks. For instance, the adversarial example "this band sucks anyways" was correctly identified as toxic by only 80.4% of respondents, a noticeably lower consensus than the other examples (the sentence with the second-lowest performance had an accuracy of 93.3%) though still comprising a clear majority. Given the high consensus for all other examples, it seems likely this discrepancy is largely due to users viewing the statement as comparatively mild and therefore perceiving it as non-toxic. Several sentences that were unambiguously non-toxic, such as "John often meets Mary", saw a human accuracy rate of 99.0%, supporting this hypothesis.

Given that human accuracy was easily maintained through all examples, we conclude that human performance is unaffected by the presence of diacritical marks even as OCR-defended model performance drops, demonstrating that OCR models perform far worse than baseline human comprehension. While our primary attack scenarios assume automated input processing where a human might not look at the text inputs at any point, given the effectiveness of the attack even with just one perturbation as shown in Figure 3 the attack could easily go unnoticed or unremarked even if the text was observed. It would be entirely feasible for an

adversary to carefully select examples with minimal perturbation that are nearly indistinguishable from the original text.

# 7 Discussion

## 7.1 Limitations

The adversarial examples in this work produce minor visual artifacts by design. These examples contrast the assumptions made by Boucher et al. (2022) in which no visual artifacts were permitted in encoding attacks. However, the results of our user study indicate that the diacritic perturbations produced do not affect the ability of humans reading text, thus giving motivation to break the imperceptibility assumption of encoding attacks.

## 7.2 Defenses

The attacks described in this paper all use Unicode diacritical marks to encode visual perturbations in the textual domain. Diacritics carry important linguistic value – particularly for pronunciation purposes – and cannot be disallowed in inputs without breaking internationalization.

However, it is reasonable to simply remove combining diacritical marks from the encoded text input to models prior to rendering for inference. In many settings, diacritical marks aren't used to encode the base meaning of language and removing them inside the model pipeline will effectively mitigate this attack without removing information necessary to the performance of the natural language model. Removing combining diacritical marks from visual text model inputs fully mitigates this attack vector, and can be accomplished by pre-processing inputs to remove all matches to the following Java-syntax RegEx: `[\u0300-\u036f]`.

In settings where diacritical marks can't be removed without hurting the performance of the model, the next best option is to perform adversarial training on the ViT, OCR, or neural encoder processing textual inputs. Yet, this will not come without additional risks via invariance-based adversarial examples where the model will learn to detect characters that are no longer human decipherable (Tramèr et al., 2020).

## 7.3 Rendering Design

In the visual text domain, setting-specific engineering details become pseudo-hyperparameters of the model. In this setting, it is no longer sufficient to communicate the model architecture and weights, but rather model providers must provide a thorough implementation or explanation of text rendering, chunking, and canvas handling along with their trained models.

One engineering challenge introduced by visual models is the need to implement text rendering onto fixed-sized canvases. By the nature of machine learning in the visual domain, model inputs must take the form of a single, fixed size image as specified by the model. This image size, which we will call the canvas dimension, introduces a key hyperparameter of the model that must be optimized. Additionally, one must also specify the font and font size that will be used for rendering. The implementation must also account for how to fit text of variable length onto the canvas, which may be further complicated if the font is not fixed width.

When there is too much text to fit onto the canvas, the canvas cannot be resized to make it fit; this would necessitate the image being further resized before inference, and the text becoming too small for the model to optimally recognize. Instead, long text inputs must be chunked into appropriately sized sections that each fit on their own canvas. These inputs must be processed separately and combined after inference (a process which must account for potential false whitespace caused by the edge of the canvas). Further, if words are broken across canvases the model will likely lose the performance benefits of recognizing adjacent characters of higher likelihood. We provide a visualization of this in Appendix A.1.

## 8 Conclusion

We have presented a novel method of attacking language models operating in the visual domain by encoding adversarial examples in the textual domain. Once rendered, these examples will generate small visual artifacts that drastically decrease model performance. We accomplish this through the injection of Unicode's combining diacritical marks via a gradient-free optimization designed to minimize target model performance over the perturbed input. We generate such adversarial examples for real-world models published by Microsoft, Facebook, IBM, and Google finding the attacks to significantly degrade model performance. We also conduct a user study which concludes that the comprehension of human users is not affected by the diacritic injections used in our attacks. In order to defend against such attacks, we conclude that model designers should remove combining diacritical marks from inputs prior to inference.

## 9 Ethics

All attacks performed as part of this work were run against local models and did not target any online services. We leveraged laboratory GPU resources for attack generation and did not use cloud services. The attack itself, however, could be applied to any external service taking in text-based user input. We notified Google of our findings on 18.05.2023 but they opted not to make changes to their transformer model, CANINE, at the time. We notified Microsoft, Meta, and IBM around the same time but have not heard back as of this writing.

Our user study was reviewed and approved by our Institutional Review Board, and no personal information was tracked from survey participants. We compensated users via Amazon Mechanical Turk at fair market prices. Since the user study involved viewing toxic content, we included a warning in the survey title that the assignment contains "language that may be considered rude or disrespectful."

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

# A  Appendix

## A.1  Rendering Design

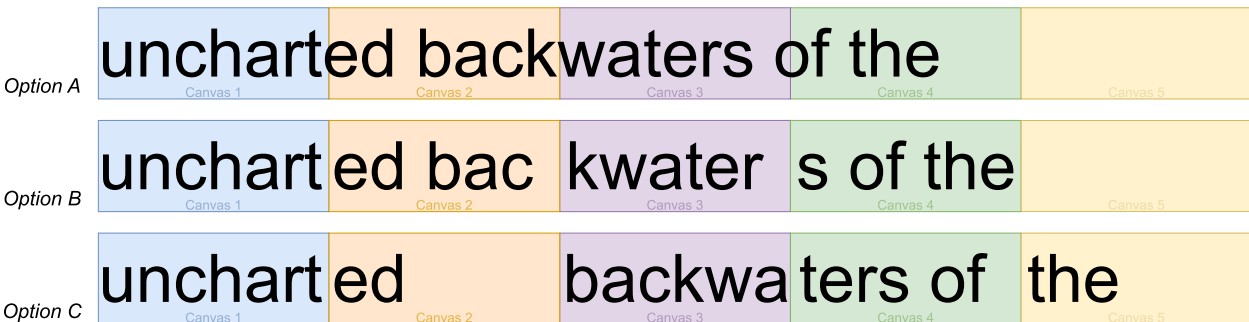

Figure 4: An example of three chunking options to render the text "uncharted backwaters of the" onto a series of fixed-size canvases for inference. Option A breaks letters across canvases, which will cause failures during inference. Option B breaks the word *backwaters* across more canvases than necessary, decreasing the model's ability to recognize likely adjacencies. Option C is likely the strongest choice in this example, but as with all other options the model pipeline must account for whitespace added by chunking after inference.

## A.2 Metric-Specific Experimental Results

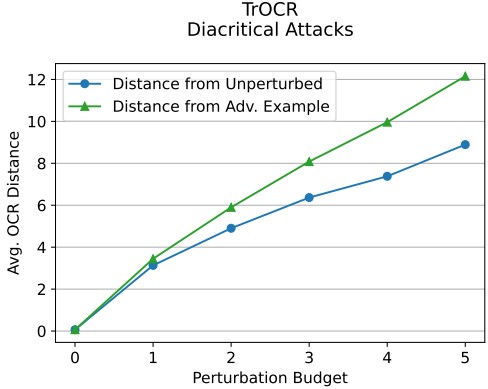

Figure 5: Evaluation of diacritic adversarial examples against TrOCR.

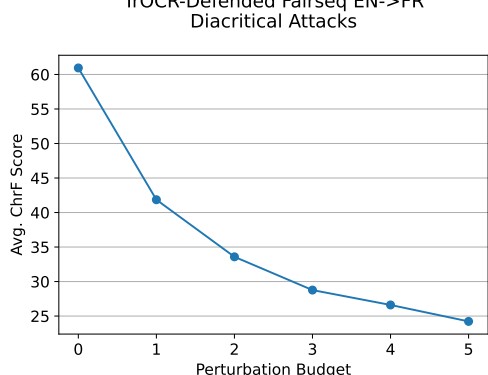

Figure 6: Eval of diacritic adv. examples against FairSeq EN->FR translation model defended by TrOCR.

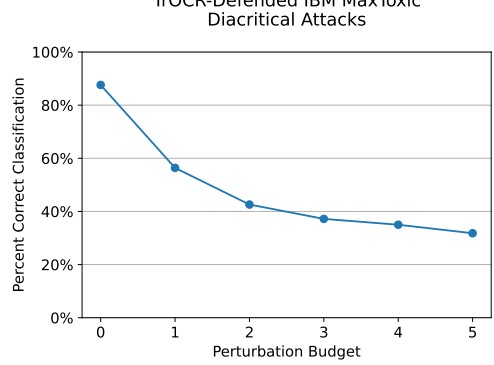

Figure 7: Eval of diacritic adv. examples against IBM's toxic content detection model defended by TrOCR.

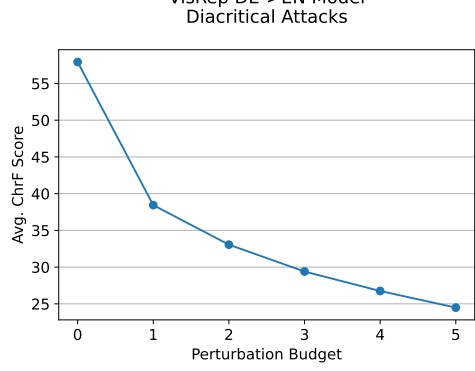

Figure 8: Eval of diacritic adv. examples against Visual Representation fork of FairSeq for DE->EN translation.

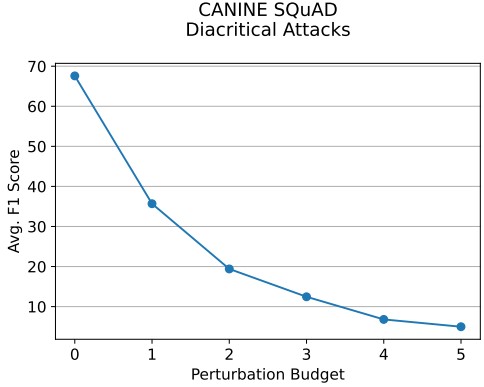

Figure 9: Eval of diacritic adv. examples against SQuAD transformer using Google's CANINE neural encoder.

## A.3 User Study Questions

A) For each of the following five sentences, retype the sentence below using only standard English characters. For example, if the sentence given is "The wåll iš red", type "The wall is red".

1. The tüb leaked water.

_______________________________________________

2. It annoys me that Fido barks.

_______________________________________________

3. I'll work on it if I can.

_______________________________________________

4. An evil thought struck Dave.

_______________________________________________

5. John lay the ball in the box.

Figure 10: First survey question measuring human readers' comprehension of text containing diacritics.

B) Toxicity is defined as "a rude, disrespectful, or unreasonable comment that is likely to make someone leave a discussion." For each of the following sentences, check the box next to the sentence if the sentence is toxic.

1. You are a coward and a fool. ☐
2. Pìç of the day—Wednesday ☐
3. It is a golden hair. ☐
4. It did not rain. ☐
5. Ahå - found it . - ☐
6. Your mother is illiterate! ☐
7. Yĵur house is ūǧly, too. ☐
8. this band sucks anyways ☐
9. John often meets Mary. ☐
10. :I'ĺl reply on your talk page. ☐
11. ::Ah yes, an asshole ☐
12. I didn't vandalize your user page, idiot. ☐

Figure 11: Second survey question measuring human readers' ability to identify toxic text containing diacritics.

