# OpenReview forum: "When Vision Fails: Text Attacks Against ViT and OCR"
_TMLR — Rejected by TMLR_

### Review · Reviewer_X6iU · 2024-10-05

**Summary Of Contributions:**

This paper introduces a novel approach for generating adversarial examples by leveraging Unicode diacritical marks to perturb input samples. The method is evaluated on various language models designed for tasks such as machine translation (MT) and toxicity classification. Results demonstrate the effectiveness of these perturbations in successfully disrupting the language models, causing them to fail in performing their intended tasks.

**Audience:**

Yes

**Claims And Evidence:**

Yes

**Requested Changes:**

- More extensive evaluation on generative tasks and LLMs would be beneficial
- Add more baselines and check how well it fares against other kinds of attacks
- Paper needs better organization and clarity

**Strengths And Weaknesses:**

### Strengths
- **Novel Perturbation Technique**: Introduces an approach for designing adversarial attacks using Unicode diacritical marks.
- **Effectiveness**: Demonstrates the ability to disrupt and cause model failures across various language tasks and thus serves as an adversarial attack

### Weaknesses
- **Poorly Written**: The paper suffers from unclear writing, impacting readability and comprehension.
- **Limited Evaluation**: The method is only tested on simple tasks and models, limiting the generalizability of the findings.
- **Lack of Baselines**: The study does not compare its approach against established baseline methods, making it difficult to assess relative effectiveness.

---

> ### Author Response · Authors · 2024-10-29
> **Author Response to Review X6iU**
>
> Thank you for your thoughtful review. We would like to offer the following responses:
> - **Poorly Written**: We note that two of the other reviewers emphasized writing quality and structure as a strength of the paper, but we acknowledge that there are certain key terms that could benefit from a clear definition earlier on within the paper to improve readability. We would be happy to revise the writing to ensure the paper is more accessible to a reader and clarify any specific sections that are unclear.
> - **Limited Evaluation**: Regrettably it is not possible to test every potential target model, but out attack vector of injection noise into inputs is generalizable across all language models, and there is not anything fundamentally different about multi-modal LLMs such as GPT-3 or GPT-4 that would impact the results. We believe that the set of target models analyzed in the paper are sufficient to prove the effectiveness of this attack, but if there are specific models that would be required to prove this efficacy, we would be happy to add them.
> - **Lack of Baselines**: For each attack, we show the performance degradation of the target model using the key metric associated with that model. To the best of our knowledge, there have not been previous diacritical mark attacks of this nature, so there is not a clear attack against which to compare beyond the effect on the target models' key metrics.

---

> > ### Comment · Reviewer_X6iU · 2024-10-31
> > **Essential Revisions Needed**
> >
> > Thank you for your responses and clarifications. I appreciate your openness to addressing areas for improvement. However, I would like to see an updated draft that more directly incorporates some of my feedback:
> >
> > - Clarity and Coherence: Please revise the draft to enhance overall clarity, coherence, and the key takeaways, making it more accessible for readers.
> >
> > - Additional Comparison: Including a comparison with state-of-the-art, open-source multimodal LLMs such as LLaVA could further strengthen the evaluation and provide additional context for your results.
> >
> > - Baselines: While direct diacritical attack baselines may not be available, any existing adversarial attack methods in the literature could serve as a useful point of comparison, even if the performance differs from your proposed method.
> >
> > An updated version incorporating these adjustments would better address the concerns and provide a stronger basis for evaluating the full impact of your work.

---

> > > ### Author Response · Authors · 2024-11-05
> > > **Implementing Revisions**
> > >
> > > Thank you for your followup comments. We are actively working on the requested edits, and will provide an update draft by November 11th incorporating all results that are available in that timeframe.

---

### Review · Reviewer_DKE8 · 2024-10-12

**Summary Of Contributions:**

This paper creates an adversarial attack on machine learning classifiers having textual input by perturbing letters with "diacritical marks" like accents (é, ô). The attack utilizes an evolutionary algorithm, and only requires access to the pre-final layer output (i.e., logit values) of the model. The attack is very successful in the settings studied, although there exists an effective defense of removing such accents as an input preprocessing step (and the authors acknowledge this).

**Audience:**

Yes

**Claims And Evidence:**

Yes

**Requested Changes:**

The following changes would improve the paper, ordered by the line number in the paper. Critical requested changes are marked with [C].

Abstract:

1. [C] Line 1: "models rely on visual inputs of rendered text as a defense strategy". This is very unclear as the first line of the paper -- What does rendering mean here? How is this a defense strategy?
2. "combining diacritical marks" -- At this stage, It is unclear what are diacritical marks, and how to combine them.

Introduction:

3. "Existing defenses ... retrofit existing models with defenses": This one sentence explanation is not understandable, it would be much better to move this to later, and explain fully.
4. "visual text domain": The following explanation is unclear, partially because "render" is not defined yet.
5. "combining marks": Undefined so far.
6. Discussion that text attacks are important because of legal applications: This seems like a stretch at this point, and is perhaps better to not mention it here. Also, what is "physical domain"?
7. [C] Overall, a significant rewrite of the introduction would be needed to get the point across: It would be good to start with briefly defining all the required concepts with examples, like diacritical marks, then state the existing defenses and what they do exactly, then say that all of them can be bypassed by this new attack, and then that this attack remains imperceptible to humans.

Background:

8. "rendered" and "encoded": It would be good to define these terms, and provide a clear example, in the main text. Currently, they are vaguely defined, and a partial visual example is provided in Figure 1.
9. [C] Adding to the confusion, in (1), the variable $t$ seems to refer to both the original text, say "hello", as well as a modified text, say "héllô". What is it exactly? Also, what does the sign $\Leftrightarrow$ mean? It would be good to provide an example of both sides of the equation, i.e., specify what is $U(t)$ and $v(t)$.

> In the following, this reviewer interpreted the manuscript by assuming that "text" $t$ refers to the underlying object / entity / concept that both "hello" and "héllô" represent, $U({\rm hello}) = \verb!\u0048! \ldots$, and $v({\rm hello})$ is the graphic "hello" that one views on a computer screen.

10. "homoglyhs": Some examples would be good here.
11. [C] Figure 2: The placement of the "adversarial example" graphic is confusing -- the reviewer was under the impression so far that the adversarial example is the initial input, but the figure seems to suggest that the adversarial example is inserted after the encoding in the OCR pipeline, and after the rendered text in the Transformer pipeline. Which one is it?
12. [C] Figure 2: Adding to (9), it would be good to start the figure with a "text $t = \ldots$" input box, then keep the encoding box free of any graphic text, then keep the rendered text as an image of this text, and so on.

13. Eq (2): What is an embedding, does this refer to a vector in $\mathbb{R}^n$? It would be good to provide an example.
14. What is a "greenfield model"? It would also be good to give an example of the "visualization gap".

Methods:

14. Discussion on Diacritics: This is a good introduction to diacritics, and would help the reader a lot if moved the the beginning of the paper.

15. [C] Main Attack Algorithm: Firstly, Algorithm 1 is the main contribution, and should be moved to the main text. Then, a discussion on computational cost (e.g., an ablation over $s, m$ vs effectiveness of the attack) is needed, it is well known that black box methods take too long to produce a reasonable output, and such a time evaluation helps the reader situate the efficacy of an attack.

16. [C] Main Attack Algorithm "logit of target class": Till here this reviewer was under the impression that only model output (class predicted) is needed for the attack, but it seems that the logit outputs are needed from the Algorithm. This is ok, but should be specified clearly in the paper.

17. [C] Main Attack Algorithm: What is "rand"? Is $|p_j| = \beta$? What is the check $R = k$ doing? There is a typo in the crossover step $p_{b_k} - p_{b_k}$. In general, several terms are not defined, and at several places, it is unclear what is the algorithm doing. An explanation is needed for every non-trivial chunk of the attack algorithm.

18. [C] Appendix A.4. Attack Examples: It seems that many of the attacks on toxicity modify the toxic word (e.g., ugly -> ûğly) -- it might be a good baseline for the toxicity example to just manually modify the toxic words by accents and compare the attack performance (Algorithm 1 should perform better, but this would potentially be much faster).

Evaluation:

19. [C] Table 1: What is "baseline performance" here? It is difficult to understand this table without context on what the performance metric of each row is. The metric-specific visualizations from the Appendix should be moved here into the main paper.

20. "chrF metric": This is non-standard and should be defined, or an intuition should be provided.

Miscellaneous:

21. Consider moving the Ethics and Fair Wage section to the end after conclusions.

**Strengths And Weaknesses:**

Strengths:
1. The paper proposes a simple unicode-modification attack that works well against undefended models. While this threat model of modifying unicode input is well known in the security and NLP community, it highlights that a seemingly benign decision of allowing an expanded set of characters (e.g., {e} vs {è, é, é, ë}) can lead to vast adversarial vulnerabilities in a machine learning pipeline.

Weaknesses:

Technical:

1. Relationship of main contribution to [Boucher+22]: The main attack Algorithm 1 seems to be very similar to Algorithm 1 in [Boucher+22], potentially because both use the same underlying black-box attack skeleton. This relationship is not commented on, and the significant differences, if any, are not emphasized.

2. Design, Computational Cost, and Context: There is a general lack of intuition behind the design decisions taken in the main algorithm, and how they affect performance of the attack. It is important to thoroughly describe the motivation of each component of the main contribution (i.e., the attack), and situate these choices amongst a wide literature on black box attacks.

3. Better Baselines needed: The preprocessing defense of (unicode -> rendering an image -> converting to text using optical character recognition -> unicode) was proposed as a simple catch-all defense, in passing just before the conclusion of [Boucher+22], and is not the main contribution of [Boucher+22] at all. Hence, it seems to be a stretch that this paper uses [Boucher+22]'s OCR defense as a major baseline -- indeed several natural steps in any serious defense, e.g., adversarial training against distorted inputs, were not tried in [Boucher+22] as a result. It is not surprising thus that this OCR defense fails against small distortions such as o -> ô.


Writing:

4. Clarity: The writing often has terms (like "diacritical marks", "rendering") without examples or introduction, making it hard to follow the manuscript. The figures have similarly under-defined terms.
5. Paper Structure: Important parts of the paper are often relegated to the Appendix (like the main Algorithm, Figures).

---

> ### Author Response · Authors · 2024-10-29
> **Author Response to Review DKE8**
>
> Thank you for your thoughtful review. We would like to offer the following responses:
> - **Relationship to [Boucher+22]**: We would be happy to expand on the description of Algorithm 1, and clarify that it uses the same optimization technique as [Boucher+22], which was originally used by [Storn&Price+97].
> - **Design, Computational Cost, and Context**: Section 3.2 explains Algorithm 1, however we would be happy to add additional context to this section to explain the optimization algorithm in a more interpretable manner.
> - **Better Baselines**: This paper leverages [Boucher+22] as a major baseline, primarily because we believe it is the closest related work. We would be happy to add comparisons against other attacks that are sufficiently similar for comparison if such exist.
> - **Clarity**: We would be happy to add formal definitions for each of the key terms to the paper, and to ensure that the terms are introduced in a way that is more understandable to the reader. We greatly appreciate the specific examples you provided to guide us in improving the readability of the paper.
> - **Paper Structure**: We would be happy to shift portions of the Appendix into the main paper, such as Algorithm 1. We had initially placed these items into the Appendix to maintain a concise length and improve readability, but we would be happy to relocate them.

---

### Review · Reviewer_RHVa · 2024-10-16

**Summary Of Contributions:**

The paper introduces a novel method for attacking visual defenses used in text-based models, demonstrating that these defenses are inadequate. Specifically, the authors exploit Unicode combining diacritical marks to manipulate text inputs, creating small visual perturbations that appear when the text is rendered, thereby corrupting the outputs of OCR and Vision Transformer (ViT) models. Additionally, the paper includes a human comprehension study to confirm that the perturbed text remains readable to humans. Experimental results show a significant drop in model performance due to this attack.

**Audience:**

Yes

**Broader Impact Concerns:**

The paper includes an ethics section stating that the proposed methods do not jeopardize any online services. However, it should also mention broader potential negative impacts. A more comprehensive discussion of these risks would enhance the ethical considerations of the work.

**Claims And Evidence:**

Yes

**Requested Changes:**

1. **Address Human Detectability and Effort** (Critical)
The paper should explore not just human readability but also the detectability of the adversarial perturbations. From the examples, it seems humans can easily spot the perturbations, which reduces the stealth of the attack. Including an analysis of human detection effort or recontextualizing the focus on automated systems (e.g., legal document scanning without human supervision) would clarify the method's real-world relevance.

2. **Reorder Related Work Section** (Non-Critical)
The related work section should be moved earlier in the paper to align with typical research paper structure, providing context for the proposed method before its introduction.

**Strengths And Weaknesses:**

### Strengths:

1. **Good Writing and Clear Figures**: The paper is well-written, with clear explanations, and the figures are easily understandable, helping to effectively convey the key concepts.

2. **Novelty in Attacking Visual Defenses**: The paper introduces a fairly novel method for bypassing traditional visual defenses used in text-based models, such as OCR and Vision Transformers. This demonstrates a significant gap in current defenses, adding value to the field of adversarial attacks.

3. **Effective Attack on Traditional Models**: The proposed method, which exploits Unicode combining diacritical marks, is shown to be highly effective. It significantly degrades the performance of the targeted models, with reported drops as high as 92.6%, showcasing the practical applicability of the attack.

4. **Experimentation**: The results section is thorough and well-supported by experiments. The evaluation provides valid and clear evidence that the attack significantly affects model performance across a variety of real-world systems.

### Weaknesses:

1. **Limited Consideration of Human Detectability**: While the paper includes a human comprehension study to ensure that perturbed text remains readable, it does not sufficiently address the detectability of these perturbations by humans or the cognitive effort required to comprehend the manipulated text. From the provided examples, it appears that humans can easily detect the perturbations, which contrasts with other traditional adversarial attacks that are designed to be imperceptible. This raises a concern about the real-world applicability of the attack in situations where humans are involved. For instance, in the case of the legal document scanning example (where documents are scanned without much human supervision), the human readability test seems less relevant. If human factors aren't a concern in such scenarios, testing human comprehension becomes less meaningful. In fact, even nonsensical characters like "$&~" could disrupt model performance, so the emphasis on readability may cloud the overall usability of the method. The paper would benefit from addressing this issue more directly.

2. **Unusual Placement of Related Work Section**: The related work section is placed just before the conclusion, which deviates from the standard structure of most research papers. Typically, related work is positioned earlier to provide context before the proposed method is introduced. The paper doesn't explain this deviation, and it would be useful to clarify the reasoning behind this structure, as it might confuse readers or disrupt the flow of the paper.

---

> ### Author Response · Authors · 2024-10-29
> **Author Response to Review RHVa**
>
> Thank you for your thoughtful review. We would like to offer the following responses:
> - **Limited Consideration of Human Detectability**: In automated scenarios with little to no human involvement (the primary attack scenario in our paper), perturbation detectability is not a concern because models are unable to distinguish between core text input and added adversarial perturbations, as demonstrated by our results. In scenarios where a human may be reviewing the text in addition to the model, we believe it is likely that humans may notice the marks but not assign them much importance (e.g., assume they are the result of a typo or printer error). We show in Figure 3 that our attack is effective even with only one perturbation (which would be nearly indistinguishable from the original text and could easily go unnoticed or unremarked), dropping performance between 25-50%. Given the variety of diacritical marks available, it would be entirely feasible for an adversary to carefully select adversarial examples that were visually less likely to be spotted. We do not believe that running an experiment specifically dedicated to detectability would yield any significant insights given the ability to launch an effective attack using only a small number of adversarial perturbations, but we will add further clarification throughout the paper text that our primary attack focus is on automated systems.
> - **Unusual Placement of Related Work Section**: The related work section was placed later in the paper in an attempt to front-load the information that was of greatest impact. While this does deviate from common editorial decisions, we've recently seen this placement advertised an (opinionated) best practice to make it faster for experts with pre-existing knowledge of related work to read papers, while maintaining the literature review for those less familiar. That being said, we would be happy to relocate this section to early in the paper if desired.

---

### Review · Reviewer_RCLJ · 2024-10-18

**Summary Of Contributions:**

This study introduces a groundbreaking method for compromising the visual defenses of text-based models, highlighting their vulnerabilities. By using Unicode combining diacritical marks, the authors subtly modify text, causing visual distortions upon rendering that disrupt the outputs of OCR and Vision Transformer (ViT) models. A human comprehension experiment confirms that, despite these alterations, the text remains readable to human users. Experimental results demonstrate a significant drop in model performance when exposed to this type of attack.

**Audience:**

Yes

**Claims And Evidence:**

Yes

**Requested Changes:**

### Expand Baseline Evaluations:
The experimental approach would be significantly strengthened by including additional baselines and exploring a wider range of settings. Evaluating the method across multiple tasks, diverse models, and different language translations would provide a more thorough analysis.

### Update to Recent Benchmarks:
The paper primarily references baselines from before 2022. To ensure a more current and comprehensive comparison, it’s important to incorporate benchmarks from 2023 and 2024, reflecting the latest advancements in the field.

### Investigate Attacks on VLMs and LLMs:
Extending the study to examine the impact of these attacks on Vision-Language Models (VLMs) and Large Language Models (LLMs) would offer valuable insights. This could open new research avenues and deepen understanding of model vulnerabilities across different architectures.

### Address Human Detectability:
While the paper focuses on human readability, it should also consider how easily the adversarial perturbations can be detected by humans. Since these manipulations appear relatively easy to spot, the discussion should explore their detectability, especially in automated scenarios where human involvement is minimal, to better assess the stealth and practical applicability of the attack.

**Strengths And Weaknesses:**

### Strength

The paper is well-structured and written in a clear and accessible manner, making it easy for readers to follow the progression of ideas and understand the technical details presented. Each section builds logically on the previous one, ensuring a cohesive flow of information throughout the study.

The research explores a novel dimension of attack by leveraging diacritical marks, a technique that has not been extensively covered in prior studies. This innovative approach introduces subtle visual perturbations into text inputs, exploiting the use of Unicode combining diacritical marks to bypass visual defenses in models. By doing so, the paper addresses an important gap in current adversarial attack strategies, offering new insights into potential vulnerabilities in text-based models.

Additionally, the paper effectively demonstrates that the attack was both successful and recognizable by human users. Through a human comprehension study, the authors validate that the perturbed text remains readable, confirming that humans can still interpret the altered text despite the visual distortions. This further solidifies the significance of the attack, as it shows that while the models are compromised, human readability is not affected, highlighting a clear discrepancy between machine and human performance under such adversarial conditions. The experimental results underscore the effectiveness of this attack method, as it leads to a notable degradation in model performance, further proving its impact.

### Weakness

The experimental approach was solid, but it lacked thorough exploration. Additional baselines and a wider range of settings need to be addressed. For instance, incorporating evaluations across multiple tasks, a variety of models, and different language translations would significantly strengthen the analysis.

While the paper touches on some potential defense mechanisms, the discussion remains quite limited. Moreover, the baselines referenced are primarily before 2022, and integrating more recent benchmarks, particularly from 2023- 2024, would provide a more current and comprehensive comparison.

It would also be valuable to investigate how these attacks perform on Vision-Language Models (VLMs) and Large Language Models (LLMs). This could open up an interesting new avenue of research, expanding the impact of the study and offering deeper insights into model vulnerabilities across different architectures.

The paper prioritizes human readability but fails to address how easily humans can detect the perturbations. In automated settings, where human involvement is minimal, readability tests seem less relevant. The detectability of these attacks, which appear easy to spot, needs more attention to assess their stealth.

---

> ### Author Response · Authors · 2024-10-29
> **Author Response to Review RCLJ**
>
> Thank you for your thoughtful review. We would like to offer the following responses:
> - **Additional Baselines**: For each attack, we show the performance degradation of the target model using the key metric associated with that model. To the best of our knowledge, there have not been previous diacritical mark attacks of this nature, so there is not a clear attack against which to compare beyond the effect on the target models' key metrics. It is not possible to test every potential target model, but we have analyzed all major targets and believe that the set of target models analyzed in the paper are sufficient to prove the effectiveness of this attack. If there are specific models that would be required to prove this efficacy, we would be happy to add them.
> - **Recent Models**: Additional models have been released and become popular while this paper has been under review, making the selection of target models something of a moving target. The models selected represented the latest advancements when this study took place. We believe that the current set of models sufficiently demonstrates the efficacy of this attack.
> - **VLMs & LLMs**: We do not believe that adding additional models would materially change the paper; we believe that the current set of models reasonably indicate the efficacy of this attack. Our attack vector of injection noise into inputs is generalizable across all language models, and there is not anything fundamentally different about multi-modal LLMs such as GPT-3 or GPT-4 that would impact the results. If the reviewers believe that specific additional models are required, however, we would be happy to add those models.
> - **Human Detectability**: In automated scenarios with little to no human involvement (the primary attack scenario in our paper), perturbation detectability is not a concern because models are unable to distinguish between core text input and added adversarial perturbations, as demonstrated by our results. In scenarios where a human may be reviewing the text in addition to the model, we believe it is likely that humans may notice the marks but not assign them much importance (e.g., assume they are the result of a typo or printer error). We show in Figure 3 that our attack is effective even with only one perturbation (which would be nearly indistinguishable from the original text and could easily go unnoticed or unremarked), dropping performance between 25-50%. Given the variety of diacritical marks available, it would be entirely feasible for an adversary to carefully select adversarial examples that were visually less likely to be spotted. We do not believe that running an experiment specifically dedicated to detectability would yield any significant insights given the ability to launch an effective attack using only a small number of adversarial perturbations, but we will add further clarification throughout the paper text that our primary attack focus is on automated systems.

---

### Author Response · Authors · 2024-10-29
**Author Combined Response to Reviews**

We would like to thank all of the reviewers for their thoughtful feedback. We have provided responses below to each review on the key points of discussion.

We would like to emphasize what we see as the main contribution of this paper: a novel attack that causes significant degradation in the performance of a variety of NLP models. We evaluate our attacks against five different target models produced by Facebook, Microsoft, IBM, and Google, showing the vulnerability of each model examined. We imagine this attack being particularly effective in scenarios such as bulk-scanned legal documents given as part of a legal discovery process.

Overall, we believe that this attack is of meaningful interest to the adversarial NLP community, whose knowledge of this attack vector could increase the level of defenses built into the next generation of models.

---

> ### Comment · Reviewer_DKE8 · 2024-10-31
> **Please make the Proposed Edits**
>
> I urge the authors to make the proposed changes in the manuscript, preferably clearly marking the edits (e.g., using a different text color). This will be essential for my re-evaluation of the paper in light of the authors' responses.

---

> > ### Author Response · Authors · 2024-11-05
> > **Implementing Revisions**
> >
> > Thank you for your followup comments. We are actively working on the requested edits, and will provide an update draft by November 11th incorporating all results that are available in that timeframe.

---

### Author Response · Authors · 2024-11-12
**Edits Incorporated into Submission**

We have uploaded a revised version of the paper for this submission incorporating as many of the major edits requested by the reviewers as we were able to make during the rebuttal timeframe. For ease of review, we have added a diff of the previously reviewed version in the Supplementary Material field.

We note that this revision does not include new experimental results due to the limited timeframe. We re-implemented a small collection of adversarial example attacks that target traditional NLP models during this rebuttal period -- which, we again note, is a different class of attacks than the visual text models we target -- to offer a partial point of reference when run against the same datasets. However, sequence-to-sequence text adversarial examples take a very long time to generate, and it will be multiple more weeks until these compute jobs would complete. We believe that these results would add limited value to the overall takeaways of the paper as they would represent an apples-to-oranges comparison of different attacks on different classes of models, so we have not attempted to include any partial results from this work in the updated revision. Some reviews additionally suggested comparing our results with other NLP attacks in the literature. We have added text in the revised submission clarifying that no other attacks targeting visual text models appear in past literature, to our knowledge, making it impossible to perform a direct comparison.

Other reviewer-requested changes are easily consumable via the diff attached as Supplementary Material. We thank the reviewers once more for their time and feedback.

---

### Decision · Action_Editor_PGra · 2024-12-27

**Recommendation:** Reject

**Comment:**

This manuscript presents a novel class of adversarial attacks using Unicode combining characters to create subtle visual perturbations that deceive machine learning models, including OCR-based defenses, without impacting human comprehension. The authors demonstrate the attack's real-world relevance through experiments on production models and support their claims with a user study showing that the perturbations remain imperceptible to human readers. The contribution is sound, highlighting a previously underexplored vulnerability in ViT and OCR.

However, several concerns remain unresolved despite the revisions. Key concepts are poorly defined, making it difficult to understand the attack's mechanics. The attack algorithm, though included in the main text, requires more detailed explanation to ensure reproducibility. Additionally, the evaluation is limited to simple tasks and lacks comparisons with established baselines or alternative attack methods, weakening the claims of generalizability and effectiveness. While the authors provided some arguments on why other baselines cannot be included during the rebuttal, it is suggested that those possibilities are *discussed* in the paper instead of hand-wavingly claiming that there are no direct baseline metrics. In addition, the differentiation from prior work, such as [Boucher+22], is insufficiently addressed. Overall, the evidence to the claims are not clear and sufficient at the current point to warrant TMLR publication.

**Audience:**

yes

**Claims And Evidence:**

not completely (mainly clarity issue)

**Resubmission Of Major Revision:**

The authors may consider submitting a major revision at a later time.